# Potential Novel N-Glycosylation Patterns Associated with the Emergence of New Genetic Variants of PRRSV-2 in the U.S

**DOI:** 10.3390/vaccines10122021

**Published:** 2022-11-26

**Authors:** Igor A. D. Paploski, Dennis N. Makau, Nakarin Pamornchainavakul, Julia P. Baker, Declan Schroeder, Albert Rovira, Kimberly VanderWaal

**Affiliations:** 1Department of Veterinary Population Medicine, University of Minnesota, St. Paul, MN 55108, USA; 2School of Biological Sciences, University of Reading, Reading RG6 6AJ, England, UK; 3Veterinary Diagnostic Laboratory, University of Minnesota, St. Paul, MN 55018, USA

**Keywords:** glycosylation, PRRSV, multi-strain dynamics, epidemiology, emergence

## Abstract

Glycosylation of proteins is a post-translational process where oligosaccharides are attached to proteins, potentially altering their folding, epitope availability, and immune recognition. In Porcine reproductive and respiratory syndrome virus-type 2 (PRRSV-2), positive selection pressure acts on amino acid sites potentially associated with immune escape through glycan shielding. Here, we describe the patterns of potential N-glycosylation sites over time and across different phylogenetic lineages of PRRSV-2 to better understand how these may contribute to patterns of coexistence and emergence of different lineages. We screened 19,179 PRRSV GP5 sequences (2004–2021) in silico for potential N-glycosylated sites. The emergence of novel combinations of N-glycosylated sites coincided with past PRRSV epidemics in the U.S. For lineage L1A, glycosylation at residues 32, 33, 44, 51, and 57 first appeared in 2012, but represented >62% of all L1A sequences by 2015, coinciding with the emergence of the L1A 1-7-4 strain that increased in prevalence from 8 to 86% of all L1A sequences from 2012 to 2015. The L1C 1-4-4 strain that emerged in 2020 also had a distinct N-glycosylation pattern (residues 32, 33, 44, and 51). From 2020 to 2021, this pattern was responsible for 44–47% of the L1C sequences, contrasting to <5% in years prior. Our findings support the hypothesis that antigenic evolution contributes to the sequential dominance of different PRRSV strains and that N-glycosylation patterns may partially account for antigenic differences amongst strains. Further studies on glycosylation and its effect on PRRSV GP5 folding are needed to further understand how glycosylation patterns shape PRRSV occurrence.

## 1. Introduction

Glycosylation of proteins is a common post-translational process in which oligosaccharides are added to specific portions of proteins [1]. These additions may impact several biological functions of proteins, including folding and indirectly modulating interactions between polypeptide chains by “masking” or “exposing” particular portions of the peptides [1,2], which, from an immunological standpoint, may modify how certain epitopes are recognized by hosts. Several types of glycosylation have been described, but the most commonly studied is N-linked glycosylation, which may occur when a specific set or sequon of amino acids (Asn-Xxx-Thr/Ser) occur in a protein [3].

The impact that glycosylation may have on human infections has been investigated under the context of several viruses’ envelope glycoproteins [4]. The interactions between glycosylated viral proteins and cellular receptors have been shown to influence viral entry in host cells by viral hepatitis [5], while N-linked glycosylation is thought to be a key component of HIV evasion from humoral immunity [6,7]. More recently, changes in the glycosylation patterns of the SARS-CoV-2 spike protein have been associated with the emergence of new variants of interest [8].

Porcine reproductive and respiratory syndrome virus-type 2 (PRRSV-2), which is an RNA virus in the *Betaarterivirus* genus and *Arteriviridae* family (PRRSV-2), is the costliest disease affecting the U.S. swine herd, with annual losses estimated to be greater than USD 650 million dollars [9]. PRRSV-2 is genetically diverse, with numerous lineages and sub-lineages described [10,11,12]. The emergence of different strains of PRRSV-2 is thought to be at least partially driven by immunological processes [12,13]. N-glycosylation is relevant to this topic because it has the potential to modify how the virus is recognized by the host immune system. The prototype VR2332 strain GP5 protein contains at least three N-glycosylation sites that are thought to be important for virus assembly and entry into host cells [14]. The main envelope protein, GP5, is a transmembrane protein that contains the Principle Neutralizing Epitope (located roughly between residues 35 and 51), which is flanked by hypervariable regions 1 and 2 (residues 32–39 and 57–61, respectively) (Figure 1) [12,15,16,17]. Residues in the hypervariable positions are frequently reported to be glycosylated [18,19] and are often under positive selection pressure [11,20]. Animals infected with PRRSV strains with distinct glycosylation patterns had different neutralizing antibody production profiles, suggesting that the presence or absence of specific glycosylation may alter the immunogenicity of the virus [19]. Even though macro-evolutionary and multi-strain dynamics of PRRSV are hypothesized to be at least partially immune-driven [12,21], little can be found describing glycosylation patterns on PRRSV over time [22], which could partially explain the emergence and turnover of PRRSV strains.

## 2. Materials and Methods

To explore the potential N-glycosylation of PRRSV in the U.S. over time, we analyzed data from the University of Minnesota Veterinary Diagnostic Laboratory from 2004 to 2021, comprised of 19,179 PRRSV ORF5 sequences (which encode the GP5 protein). ORF5 sequences were translated to amino acids using AliView [27]. Sequons in the translated amino acid sequence were identified using a custom-built script coded in Stata 15 [28]-code is available in Appendix A. Our code identified sequons with the pattern Asn-Xxx-(Thr/Ser), where the middle position could be any amino acid except proline, since a sequon consisting of Asn-Pro-(Thr/Ser) is not adequate for an N-glycosylation to occur [29]. Sequences were then summarized according to which residues were potentially N-glycosylated. Specific N-glycosylated residues were referred to by the position of the initial asparagine within the GP5 protein. Sequences were stratified according to the year collected and to which phylogenetic lineage or sub-lineage they belonged [10,11,12,30]. To visualize phylogenetic relationships, a time-scaled phylogenetic (ML; GTR + G) tree was constructed for 500 randomly selected sequences using Mega and TimeTree [31,32] and illustrated using Microreact [33]. Figures illustrating the percentage of sequences according to their potential N-glycosylation pattern per year per (sub-)lineage were constructed using Stata 15 [28]. Glycosylation patterns that occurred in <5% of the sequences of a given lineage or sub-lineage were grouped into “other patterns”.

## 3. Results and Discussion

Potential N-glycosylation of nine residues was identified (residues 30, 32, 33, 34, 35, 44, 51, 57, and 59), with different combinations of glycosylated sites being prevalent over time (Figure 2A). Even though the glycosylation pattern of a sequence is not necessarily lineage-defining, the emergence of certain constellations of potential N-glycosylations coincides with past PRRSV epidemics in the U.S., as can be seen in Figure 3 for L1A, L1C, and L1H. Each of these lineages had been present at low frequencies for some time (Figure 2B), but a rapid increase in frequency was associated with a novel glycosylation pattern. For L1A, sequences glycosylated at sites 32, 33, 44, 51, and 57 first appeared in 2012, and by 2014 it represented more than 40% of all L1A sequences identified, reaching a peak of 62% of all L1A sequences identified in 2015. This coincides with the emergence of the L1A 1-7-4 virus [11], which was a widely recognized event of clinical significance in the industry [34].

The emergence of the L1C 1-4-4 variant [35] in 2020 can also be observed from a glycosylation pattern perspective. This virus’s glycosylation pattern (at residues 32, 33, 44, and 51) was circulating since at least 2007, though represented only a small percentage of sequences (Figure 3-L1C). In 2020 and 2021, this pattern was observed in 44 and 47% of the L1C sequences identified in those years. More generally, the prevalence of different patterns in L1C appears to be wave-like, with different patterns prevalent at various points in time. Other lineages also display a change in the relative frequency of occurrence of glycosylation patterns over time, but those are harder to trace back to well-defined PRRSV epidemics. Glycosylation patterns are not solely responsible for the emergence of new strains, as evidenced by the fact that the pattern associated with the L1C-1-4-4 variant was present in this lineage since at least 2007. However, in a dynamic landscape of cross-immunity elicited by natural occurrence of PRRSV, diverse immunization practices through live virus exposure, and vaccination, the relative fitness of viruses with specific glycosylation patterns may change over time.

One limitation of this study is that we are only reporting sites that are potentially N-glycosylated. No spectrometric and biochemical studies were performed to evaluate if these sites are indeed glycosylated. Another limitation is that we used secondary data to conduct the analysis. The UMN VDL dataset is based upon sequencing requested by clients, introducing geographical biases (data comes from more than nine states, but 82% of the sequences are from the Midwest U.S.) and potential biases in PRRSV detection associated with reasons for sequencing (such as atypical clinical presentation). However, we believe the breadth of data available from the UMN VDL more than likely offsets this limitation. Finally, as an observational study, we cannot assess causality (is a given glycosylation pattern the cause of frequency increases for a given strain or is it just a hitchhiker present in a successful strain?). However, experimental studies show that glycosylation may play a role in protein folding [36], immune evasion [18] and recognition of epitopes and subsequent neutralization by the immune system [22], which supports the hypothesis that glycosylation pattern may be a relevant aspect when it comes to understanding the sequential dominance of different PRRSV strains in the field.

In this paper we reported trends of occurrence of potential N-glycosylation in PRRS viruses obtained from the University of Minnesota Veterinary Diagnostic Laboratory for different PRRSV lineages over different years. Protein glycosylation has been implicated in many aspects of adaptive immune response [37]. As such, the wealth of historical data available for PRRSV, with thousands of PRRSV ORF5 sequences available, should not be understated. Further exploration of glycosylation patterns, how they affect PRRSV GP5 folding, and experimental studies could help better understand how glycosylation patterns shape PRRSV occurrence and how that can be leveraged to improve PRRSV control.

## Figures and Tables

**Figure 1 vaccines-10-02021-f001:**
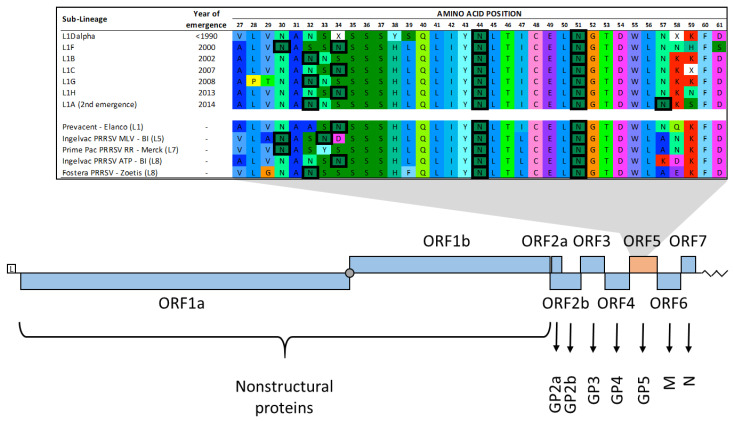
Amino acid sites within the ectodomain of GP5 with potential N-glycosylation highlighted in hyper-variable regions 1 (sites 32–39) and 2 (sites 57–61) in the ORF5 portion of the genome (**top**). Displayed examples of potential N-glycosylation are shown on the consensus sequences of sub-lineages of PRRSV that dominated the PRRSV landscape, ordered according to their year of emergence and on vaccine sequences [12]. PRRSV genome organization and expression (**bottom**): L—leader sequence (5’ UTR); filled circle—ORF 1a and 1b ribosomal frameshift site; zigzag line—3’s UTR and poly(A) tail. For additional detailed illustrations on the PRRSV genome, please refer to [23,24,25,26]. Figures partially adapted from [12,23].

**Figure 2 vaccines-10-02021-f002:**
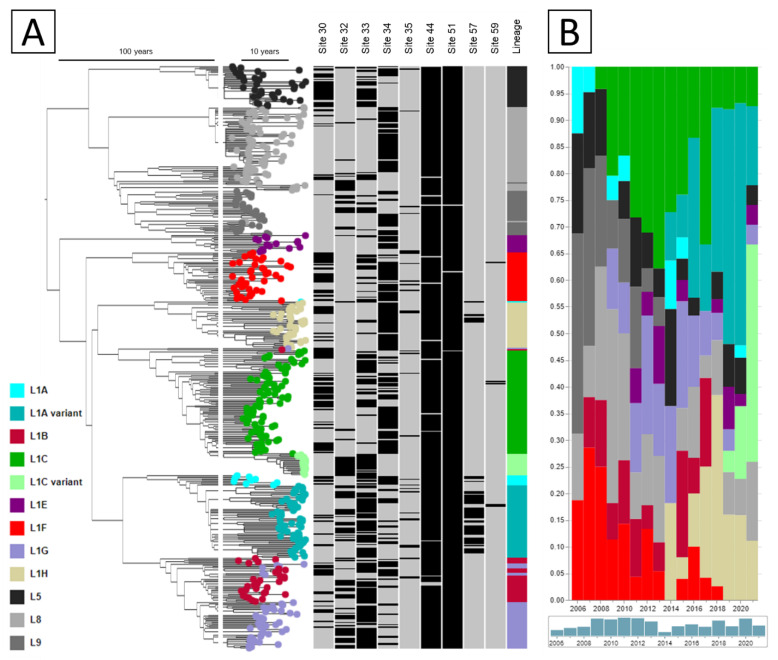
(**A**) Time-scaled phylogenetic tree of 500 randomly selected sequences illustrating if each sequence was potentially N-glycosylated (black) or not (gray) at each residue site in which glycosylation was identified. (**B**) Fraction of sequences belonging to each lineage identified in each year.

**Figure 3 vaccines-10-02021-f003:**
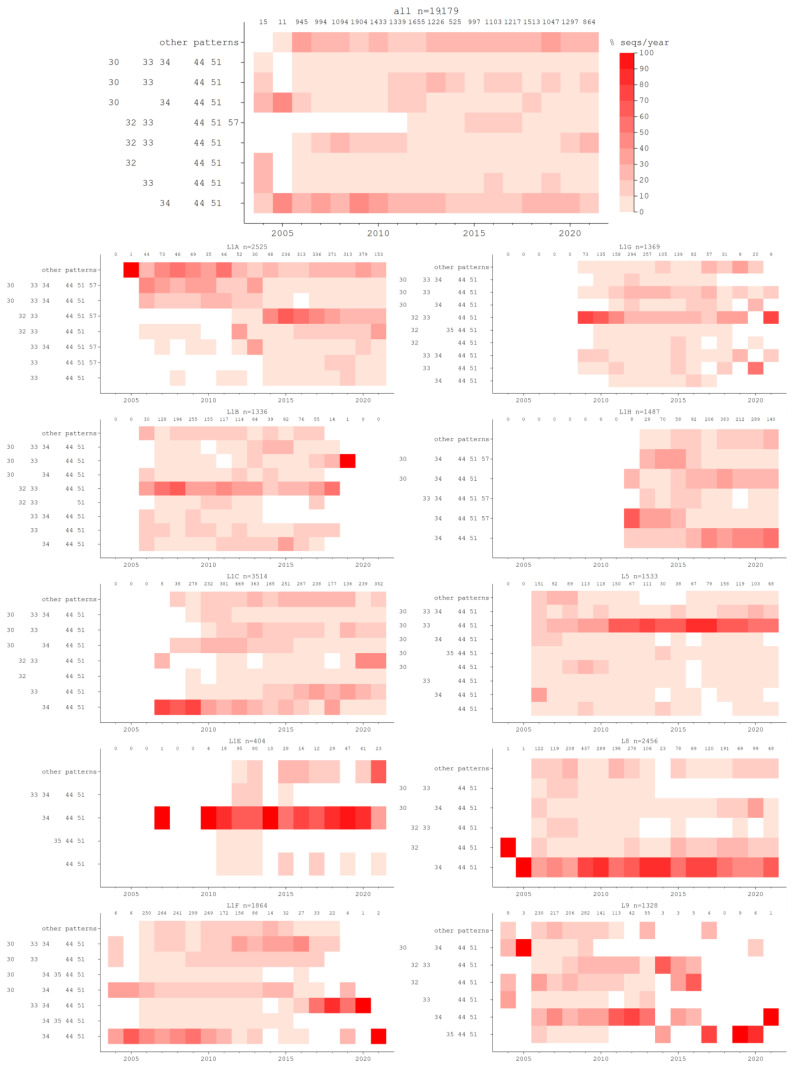
Percentage of sequences identified in each year according to their pattern of potentially N-glycosylated sites across lineages and sub-lineages. Each graph corresponds to a sub-lineage (specified on top of the graph), followed by the total number of sequences of that sub-lineage identified in each year.

## Data Availability

The data that support the findings of this study are available the University of Minnesota Veterinary Diagnostic Laboratory. Restrictions apply to the availability of these data, which were used under license for this study.

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
