# Peer review of "Potential Novel N-Glycosylation Patterns Associated with the Emergence of New Genetic Variants of PRRSV-2 in the U.S"

_vaccines, 2022, doi:10.3390/vaccines10122021_

Round 1

Reviewer 1 Report

The present brief report is a nice piece of work. the work is both important and interesting. Before acceptance in journal i think author of draft should address following points.

1) Overall draft is well written but need minor english editing here and there.

2) I think it will be good if author include brief background about virus genome compsition, ORF, proteins in brief. Will be great if same can be shown in cartoon presentation form.

3) I think author should also include amino acid sequence of reference GP5 protein and highlight the glycosylation sites (in figure form)

4) Author should also mention which version of software was used of analysis

Author Response

Reviewer 1

The present brief report is a nice piece of work. the work is both important and interesting. Before acceptance in journal i think author of draft should address following points.

1) Overall draft is well written but need minor english editing here and there.

We appreciate the comment. The entire document went through an electronic grammar correction and the authors that are native English speakers reviewed it as well.

2) I think it will be good if author include brief background about virus genome compsition, ORF, proteins in brief. Will be great if same can be shown in cartoon presentation form.

We appreciate the suggestion. We assembled a new figure that illustrates the virus genome compositions, ORFs and proteins. We highlight specifically the ORF5 portion of the genome, which is the portion that we focus our analysis of potential glycosylation. This is now Figure 1 on the paper.

3) I think author should also include amino acid sequence of reference GP5 protein and highlight the glycosylation sites (in figure form)

Added on the same figure as the comment above.

4) Author should also mention which version of software was used of analysis

We added the reference for Stata 15.

Reviewer 2 Report

I have read with great interest the results of the epidemiological study presented by the authors in the article entitled “Novel N-glycosylation patterns are associated with emergence of new genetic variants of PRRSV-2 in the U.S.” In the course of observational studies, the authors noticed some regularity in the occurrence of various combinations of N-glycosylated sites with the occurrence of successive waves of the disease in more than 9 US states. Globally, PRRS is the most economically important infectious disease of pigs. In the US alone, losses from disease outbreaks are estimated at approximately $ 700 million annually. The PRRSV-1 and PRRSV-2 genome is highly diverse, rapidly evolving but amenable to the generation of many mutants. Mutations, recombination and post -translational processes may lead to the formation of strains with different biological characteristics. This variability poses a challenge to the host's immune system and to us fighting the disease in different countries.

Personally, I really appreciate that despite the fact that the topic is relatively familiar - RNA viruses usually mutate easily and quickly, and post-translational processes are very variable - emphasized how important this problem is. When dealing with field vets, I know that this obvious thing is often forgotten.

Author Response

I have read with great interest the results of the epidemiological study presented by the authors in the article entitled “Novel N-glycosylation patterns are associated with emergence of new genetic variants of PRRSV-2 in the U.S.” In the course of observational studies, the authors noticed some regularity in the occurrence of various combinations of N-glycosylated sites with the occurrence of successive waves of the disease in more than 9 US states. Globally, PRRS is the most economically important infectious disease of pigs. In the US alone, losses from disease outbreaks are estimated at approximately $ 700 million annually. The PRRSV-1 and PRRSV-2 genome is highly diverse, rapidly evolving but amenable to the generation of many mutants. Mutations, recombination and post -translational processes may lead to the formation of strains with different biological characteristics. This variability poses a challenge to the host's immune system and to us fighting the disease in different countries.

Personally, I really appreciate that despite the fact that the topic is relatively familiar - RNA viruses usually mutate easily and quickly, and post-translational processes are very variable - emphasized how important this problem is. When dealing with field vets, I know that this obvious thing is often forgotten.

We appreciate the kind comments by the reviewer.

Reviewer 3 Report

The authors propose that novel N-glycosylation patterns are associated with the emergence of new genetic variants of Porcine Reproductive and Respiratory Syndrome Virus Type 2 (PRRSV-2). They have analyzed existing sequences of PRRSV-2 GP5 in silico.

This is a sole bioinformatic study, that is predictive in nature. The authors have nicely outlined the limitations of their study in the discussion. However, the terminology with clear distinction of a potential glycosylation site vs. the actual PTM should follow suit: Throughout their manuscript, the authors confuse identification of a potential glycosylation site, or sequon, with an actual N-glycosylation as a posttranslational modification (PTM) that occurs on an asparagine residue. Those are not the same. Just because an Asn-Xxx-Ser/Thr sequence is detected, doesn’t mean that a glycosylation has occurred on the amino acid level. If the sequence is not exposed to the accessible protein surface, or otherwise obstructed, N-glycosylation may not occur. Therefore at several occurrences the wording has to be corrected; For example: “glycosylation at residue..”, “Glycosylation of nine residues ….. was identified” (also correct grammar on that one). The term “glycosylation pattern” is also inadequate for the observed “sequence pattern of potential glycosylation sites”.

When referring to a specific residue number, do the authors mean the asparagine? Please clarify in the manuscript.

Other points for improvement:

Figure 2. font is too small and to fine, not readable.

The STATA 15 script should be shown in the supplementary data or in the manuscript.

How was proline handled in the glycosylation sequon? A short sentence on what rules were used by the STATA script to identify and score potential glycosylation sites would improve the paper.

A few more introductory sentences on PRRSV-2 and the disease it causes would be appreciated in the introduction.

Please correct typos/grammar:

Line 24: the word “of” is missing after 47%

Line 79: wrong grammar “…nine residues of was were identified…”

Author Response

The authors propose that novel N-glycosylation patterns are associated with the emergence of new genetic variants of Porcine Reproductive and Respiratory Syndrome Virus Type 2 (PRRSV-2). They have analyzed existing sequences of PRRSV-2 GP5 in silico.

This is a sole bioinformatic study, that is predictive in nature. The authors have nicely outlined the limitations of their study in the discussion. However, the terminology with clear distinction of a potential glycosylation site vs. the actual PTM should follow suit: Throughout their manuscript, the authors confuse identification of a potential glycosylation site, or sequon, with an actual N-glycosylation as a posttranslational modification (PTM) that occurs on an asparagine residue. Those are not the same. Just because an Asn-Xxx-Ser/Thr sequence is detected, doesn’t mean that a glycosylation has occurred on the amino acid level. If the sequence is not exposed to the accessible protein surface, or otherwise obstructed, N-glycosylation may not occur. Therefore at several occurrences the wording has to be corrected; For example: “glycosylation at residue..”, “Glycosylation of nine residues ….. was identified” (also correct grammar on that one). The term “glycosylation pattern” is also inadequate for the observed “sequence pattern of potential glycosylation sites”.

We appreciate and agree with the comment. We reviewed the paper to ensure it reads “potential N-glycosylation” whenever appropriate. We revisited our limitations section to ensure that the first limitation mentioned is that we are only detecting potential N-glycosylation and not evaluating if those are present via spectrometric or biochemical analysis. We expect that with those changes (regular use of “potential N-glycosylation and early acknowledgement on the limitations) appropriately convey the message to readers.

Regarding the terminology of “glycosylation pattern”, we replaced it by “pattern of potentially N-glycosylated sites” the first time they appear on the paper, but not on all instances, to avoid wordiness.

When referring to a specific residue number, do the authors mean the asparagine? Please clarify in the manuscript.

Yes. We added a sentence to clarify this point. The sentence reads “Specific potentially N-glycosylated residues were referred to as the asparagine position given that this formed a sequon on the GP5 protein.

Other points for improvement:

Figure 2. font is too small and to fine, not readable.

We replaced the file for this figure by a higher quality one and increased the size of the image.

The STATA 15 script should be shown in the supplementary data or in the manuscript.

We appreciate the suggestion. The code used to flag potential N-glycosylation sites is now available as a supplementary file. This is also now mentioned on the text.

How was proline handled in the glycosylation sequon? A short sentence on what rules were used by the STATA script to identify and score potential glycosylation sites would improve the paper.

We appreciate the attention to detail. When X is P (proline) in a sequon - NP(S/T), our code does not flag it as a potential N-glycosylation. We added a sentence to clarify this. The sentence reads “Sequons in the translated amino acid sequence were identified using a custom-built script coded in Stata 15 [24]. Our code specifically excluded sequons whose middle amino is a proline, since a sequon in the format NP(S/T) is not adequate for an N-glycosylation to occur [25].

A few more introductory sentences on PRRSV-2 and the disease it causes would be appreciated in the introduction.

We added extra introductory sentences to the third paragraph of the introduction covering the economic impact that PRRSV has in the US swine industry and why we think that N-glycosylation is potentially associated with the virus’ diversity.

Please correct typos/grammar:

Line 24: the word “of” is missing after 47%

Line 79: wrong grammar “…nine residues of was were identified…”

We appreciate the reviewer pointing to both issues. They are now corrected.